# Opposing Spatially Segregated Function of Endogenous GDNF-RET Signaling in Cocaine Addiction

**DOI:** 10.3390/biom13050761

**Published:** 2023-04-27

**Authors:** Daniel R. Garton, Giorgio Turconi, Vilma Iivanainen, Jaan-Olle Andressoo

**Affiliations:** 1Department of Pharmacology, Faculty of Medicine, Helsinki Institute of Life Science, University of Helsinki, 00290 Helsinki, Finlandvilma.iivanainen@helsinki.fi (V.I.); 2Division of Neurogeriatrics, Department of Neurobiology, Care Science and Society (NVS), Karolinska Institutet, 17177 Stockholm, Sweden

**Keywords:** addiction, dopamine, GDNF, GFRa1, RET, BDNF, nucleus accumbens, ventral tegmental area, cocaine

## Abstract

Cocaine addiction is a serious condition with potentially lethal complications and no current pharmacological approaches towards treatment. Perturbations of the mesolimbic dopamine system are crucial to the establishment of cocaine-induced conditioned place preference and reward. As a potent neurotrophic factor modulating the function of dopamine neurons, glial cell line-derived neurotrophic factor (GDNF) acting through its receptor RET on dopamine neurons may provide a novel therapeutic avenue towards psychostimulant addiction. However, current knowledge on endogenous GDNF and RET function after the onset of addiction is scarce. Here, we utilized a conditional knockout approach to reduce the expression of the GDNF receptor tyrosine kinase RET from dopamine neurons in the ventral tegmental area (VTA) after the onset of cocaine-induced conditioned place preference. Similarly, after establishing cocaine-induced conditioned place preference, we studied the effect of conditionally reducing GDNF in the ventral striatum nucleus accumbens (NAc), the target of mesolimbic dopaminergic innervation. We find that the reduction of RET within the VTA hastens cocaine-induced conditioned place preference extinction and reduces reinstatement, while the reduction of GDNF within the NAc does the opposite: prolongs cocaine-induced conditioned place preference and increases preference during reinstatement. In addition, the brain-derived neurotrophic factor (BDNF) was increased and key dopamine-related genes were reduced in the GDNF cKO mutant animals after cocaine administration. Thus, RET antagonism in the VTA coupled with intact or enhanced accumbal GDNF function may provide a new approach towards cocaine addiction treatment.

## 1. Introduction

Cocaine is one of the most commonly consumed drugs of abuse worldwide [1] with potentially severe complications including damage to the cardiovascular system and death [2]. As a competitive inhibitor of the dopamine reuptake transporter (DAT), cocaine can increase presynaptic dopamine levels in the nucleus accumbens (NAc) leading to reinforcement and forming persistent drug-context associations [3]. Indeed, mice with a knock in mutation resulting in a cocaine-insensitive DAT had abolished cocaine reward [4]. Other drugs of abuse such as opioids and alcohol have pharmaceutical treatments available for managing addiction [5,6]. Cocaine, however, has no FDA approved medications for the treatment of cocaine use disorder [7,8], highlighting a need for new therapeutic targets.

Conditioned place preference (CPP) is an important test for assessing the rewarding effects of drugs of abuse [9,10,11]. CPP can model the context-associated reinforcement from drugs of abuse including stimulants such as amphetamine and cocaine [12,13,14,15]. Delayed CPP extinction following cocaine administration and CPP reinstatement is believed to model human drug-seeking behavior and substance abuse relapse [13]. Supporting this idea, stimulant-induced CPP has been observed in rodents [15,16,17], primates [18], and humans [19].

Glial cell line-derived neurotrophic factor (GDNF) is potentially the strongest dopamine function promoting secreted neurotrophic factor [20,21,22]. In the brain, GDNF has its highest expression levels in the striatum, where it is specifically expressed in parvalbumin positive and cholinergic striatal interneurons [23,24,25,26,27]. Its receptor RET is highly expressed in the ventral tegmental area (VTA) on dopamine (DA) neurons which project to the nucleus accumbens (NAc), part of the ventral striatum which regulates reward and addiction [24,28,29,30,31] (Figure 1A). GDNF dimerizes and first binds to its binding receptor GFRa1, whereupon the GDNF-GFRa1 complex then binds to transmembrane receptor tyrosine kinase RET which activates downstream signaling in dopamine neurons [32,33,34,35,36]. GDNF, acting through its receptor tyrosine kinase RET, has been shown to activate the mitogen-activated protein kinase (MAPK) pathway and directly influence the excitability of cultured midbrain DA neurons [37]. In addition, both endogenous GDNF upregulation and downregulation lead to increased DAT activity in vivo [38,39,40]. In addition to the MAPK pathway, RET has been known to activate other signaling pathways, including the phosphatidylinositol-3-kinase (PI3K)/protein kinase B (AKT) pathway, the phospholipase C-gamma (PLCγ) pathway, and the c-Jun *N*-terminal kinase (JNK) pathway [32,41].

Mice with constitutively active RET elicited via mimicking the Met918Thr mutation from multiple endocrine neoplasia type B (MEN2B) [42,43] display severely delayed CPP extinction from amphetamine [44], suggesting that GDNF-RET signaling is an important modulator of addiction to amphetamine. In contrast with constitutively active RET, which prolongs addiction to amphetamine [44], chronic infusion of GDNF in adult animals directly into the VTA reduced cocaine-induced place preference [45] and ectopic GDNF overexpression in the NAc reduced alcohol-seeking behavior [46]. Conversely, heterozygous knockout of GDNF or chronic infusion of an anti-GDNF antibody into the VTA enhanced cocaine-induced CPP and locomotor sensitization [45].

Taken together, current data suggest that GDNF-GFRa1-RET signaling is an important modulator of addiction to commonly used drugs of abuse and may therefore prove to be a viable therapeutic approach to addiction treatment [47]. In line with this, adult onset 50% reduction in striatal GDNF levels in GDNF conditional knockout animals implemented via striatal AAV-Cre delivery displayed dampened amphetamine responses [40]. In order to address the effects of GDNF-RET signaling after the onset of cocaine addiction, here we utilize two conditional knockout mouse lines to reduce RET [48] and GDNF levels [49] in the VTA and in the NAc, respectively, using adeno-associated viral (AAV) delivery of Cre-recombinase after establishing cocaine-mediated CPP. We find that conditional 70% reduction, approximately, of RET from the VTA hastens CPP extinction and attenuates CPP reinstatement. On the contrary, about 70% GDNF reduction from the NAc prolongs CPP extinction, enhances CPP reinstatement, increases BDNF and reduces dopamine-related gene expression. Thus, reducing GDNF in the NAc and RET levels in the VTA have opposing roles in cocaine addiction. This information is important in guiding future development of GDNF-RET system modulators for treating cocaine addiction.

## 2. Materials and Methods

### 2.1. Animals

Mouse husbandry was performed as described previously [38]. Briefly, all animal experiments were carried out according to the European Union Directive 86/609/EEC and were approved by the County Administrative Board of Southern Finland (license number ESAVI/12046/04.10.07/2017). Efforts were made to minimize the number and suffering of animals. Mice were maintained in a 129Ola/ICR/C57bl6 mixed genetic background, and male mice were used for experiments with wild-type littermates used as controls. GDNF cKO mice were generated in-house [49], and RET cKO mice [48] were imported from Jackson Laboratories (JAX stock #028548). Individually ventilated cages in a specific pathogen-free environment where mice had ad libitum access to food and water were used to group house the animals. The mice were kept under a 12-h light-dark cycle (lights on at 6 a.m.) at relative humidity of 50–60% and room temperature 21 ± 1 °C. Each week, the nest material (Tapvei) and bedding (aspen chips, Tapvei) were changed. Wooden blocks (Tapvei) were provided in each cage for enrichment.

### 2.2. Genotyping

Genotyping was performed as described previously [49]. Genotyping samples were collected at weaning and/or during dissection. Ear or tail samples were routinely collected for DNA analysis. Extracta DNA Prep for PCR-tissue (Quanta Biosciences, Gaithersburg, MD, USA) was used to isolate genomic DNA from each mouse. Genotyping of each animal was performed using AccuStart II GelTrack PCR SuperMix (Quanta Biosciences, USA). After PCR, samples were analyzed on a 2% agarose gel by gel electrophoresis in tris-acetic acid-EDTA (TAE) buffer (Elatus Media Kitchen, University of Helsinki). Genotyping primers used are summarized in Table 1.

### 2.3. Intracranial AAV-Cre Injections

Intrastriatal AAV5-cre injections were performed as described previously [38]. Briefly, adult male mice were anaesthetized with isoflurane in 100% oxygen (3–4% for induction and 2% for maintenance; Oriola, Espoo, Finland), and the top of the head was shaved and fixed into a stereotaxic surgery frame. Upon stable placement of the skull, the top of the head was sterilized with Desinfektol P (Berner Pro, Helsinki, Finland), locally anaesthetized with lidocaine (Yliopiston Apteekki, Helsinki, Finland), and, using a scalpel, opened to reveal the skull. The skull was drilled into bilaterally for both NAc and VTA injections. For VTA injections, coordinates used were AP: −3.2 mm; ML: +0.5 and −0.5 mm; and DV: −4.2 mm. Total volume injected was 1 μL/hemisphere (2 μL in total) at a flow rate of 0.2 μL/min, letting needle sit after for 4–5 min (10 min total). For NAc injections, coordinates used were A/P 1.18, M/L ±1, D/V −5.0. Total volume injected was 3 μL (1.5 μL/hemisphere), at a flow rate of 0.2 μL/min, letting needle sit afterwards for 4–5 min (12 min). The virus stock was diluted with Dulbecco’s PBS 1:5 (5 μL virus and 20 μL of DPBS), with the viral titer approximately 3.3 × 10^14^ VG/mL. 

Once the injection was completed, the skin above the skull was sutured closed and the animals were administered carprofen (5 mg/kg) (Yliopiston Apteekki, Finland) as an analgesic. The next day, sutures were checked to ensure proper wound closure and healing. In the following days, the animals were monitored to ensure proper recovery, and kept for at least 2 days prior to beginning experiments.

### 2.4. Conditioned Place Preference

Conditioned place preference procedures were carried out as described previously [11,44,50]. CPP was designed using an unbiased approach [51]. In short, the CPP paradigm consists of three different phases: habituation, conditioning, and extinction. During the habituation phase, mice were allowed to explore the two compartments unhindered for 15 min. During the conditioning phase, mice were treated over 8 days in 2 blocks of 4 consecutive days with a 2-day intermission with alternating injections of cocaine at a dose of 15 mg/kg i.p. (Yliopiston Apteekki) or saline and confined into the corresponding compartment immediately after injection for 30 min. Group 1 and Group 2 animals were distinguished based on the chamber to which the cocaine stimulus was paired. Group 1 animals had cocaine administration paired to the room containing the grid-like floor pattern, whereas Group 2 animals had cocaine administration paired to the room containing the hole-like floor pattern.

Conditioned place preference was then evaluated the following day with the mice allowed to explore both compartments for 15 min. Preference was scored as the difference between time spent on the cocaine-paired side minus the group average time mice spent on the saline-paired side to reduce bias. After preference was established, mice were intracranially injected with AAV2/5-Cre developed previously [38] over 2 days. Next, for the extinction tests, the mice were allowed to explore the whole apparatus without any treatment for 15 min. RET cKO mice were then evaluated 3, 5, 7, 9, and 11 days after AAV injections. Then, two days later, CPP reinstatement with half the dose of cocaine (7.5 mg/kg) was evaluated. GDNF cKO mice were evaluated for CPP extinction over the course of a month: 4, 7, 10, 14, 17, 23, and 30 days after AAV injections. Then, CPP reinstatement was evaluated with half of the original dose of cocaine at 7.5 mg/kg 34 days after AAV injections. After this reinstatement, CPP extinction was again evaluated 36, 41, 49, and 56 days after the original AAV injections. Finally, mice were reevaluated for CPP reinstatement with half the original dose of cocaine (7.5 mg/kg) again on day 57 after AAV injections.

### 2.5. Dissections

Animals were deeply anaesthetized with CO_2_ and/or cervically dislocated, followed immediately by decapitation. The skull was quickly cut open and the brain removed and placed into ice-cold PBS prior to tissue isolation. Brain structures of interest were then dissected from 2 mm thick slices using a brain matrix (Stoetling) and immediately snap frozen using dry ice and stored at −80 °C until use for RNA extraction.

### 2.6. Reverse Transcription and Quantitative PCR

Reverse transcription and quantitative PCR (qPCR) was performed as described previously [38]. Briefly, RNA was extracted and isolated using TRIzol reagent (Invitrogen, Waltham, MA, USA) according to the manufacturer’s instructions. Each sample containing 200 ng total RNA was treated with DNase I (Thermo Fisher Scientific, Waltham, MA, USA), which was RNase-free. By incubating for 10 min with 5 mM EDTA at 65 °C, DNase I was rendered inactive. After DNase I inactivation, reverse transcription using random hexamer primers was done using RevertAid Reverse Transcriptase (Thermo Fisher Scientific). This then generated complementary DNA (cDNA). Next, the cDNA was diluted 1:10 and subsequently stored at −20 °C until needed for qPCR. The BioRad C1000 Touch Thermal Cycler upgraded to CFX384 System (BioRad, Hercules, CA, USA), was used for qPCR analysis. Each sample was analyzed in duplicates on 384-well plates. Into each well was pipetted SYBR Green I Master (Roche, Basel, Switzerland) and 250 pmol primers for 10 μL total volume. Each reaction included cDNA or a negative control (minus-reverse transcription control or water). The reference gene used for all qPCR samples was mouse ActinB. Results for a biological repeat were discarded when the C_q_ value for one or more of the sample duplicates was 40 or 0, or when the C_q_ difference between replicates was >1. Primer sequences used are depicted in Table 2.

### 2.7. Statistical Analysis

All statistical analyses were performed using GraphPad prism software. Any potential outliers were identified by Grubbs’ test with an alpha = 0.05 and removed prior to subsequent analysis. Behavioral data with repeated measures was analyzed using 2-way repeated measured ANOVA, or mixed-effects analysis if values were missing, followed by post-hoc Holm–Šídák tests to ascertain significant effects. Gene expression changes at the mRNA level and non-repeated behavioral data were analyzed using *t*-tests with Welch’s correction. Gene expression changes relative to wild type where multiple genes, and thus multiple hypotheses, were tested at once were determined using multiple *t*-tests with a single, pooled variance followed by multiple comparison correction by controlling for the false discovery rate with the Benjamini, Krieger, and Yekutieli two-stage set-up method such that a rate of 5% (Q = 5) was accepted for determining significance.

## 3. Results

### 3.1. RET cKO Mice upon AAV-Cre Delivery Have about 60–70% Reduction of RET from the VTA

Clinical management of cocaine use disorder requires interventions after cocaine addiction has previously been established. To this end, cocaine-induced conditioned place preference was first established prior to reducing RET expression in the mesolimbic dopamine pathway of RET conditional knockout (cKO) mice [48] via intra-VTA injections of AAV-Cre virus (Figure 1A). Analysis of VTA Ret mRNA expression at the endpoint (Figure 1A) revealed on average about a 60–70% reduction (Figure 1B), and thus is not a full knockout of total RET function. Prior to conditional reduction in Ret mRNA levels, the animals in group 1 and group 2 displayed differing preferences for the two chambers during habituation, with significant differences between the groups observed in both the wt/wt and the cKO/cKO animals (Figure 1C). The difference in habituation between groups 1 and 2 may reflect an inherent preference of the mice for the room with the grid-type floor pattern. However, there were no significant differences between the genotypes, as expected, given both groups have normal RET expression during this stage [48]. Preference itself also was not different between the genotypes in either group 1 or group 2 (Figure 1D,E).

**Figure 1 biomolecules-13-00761-f001:**
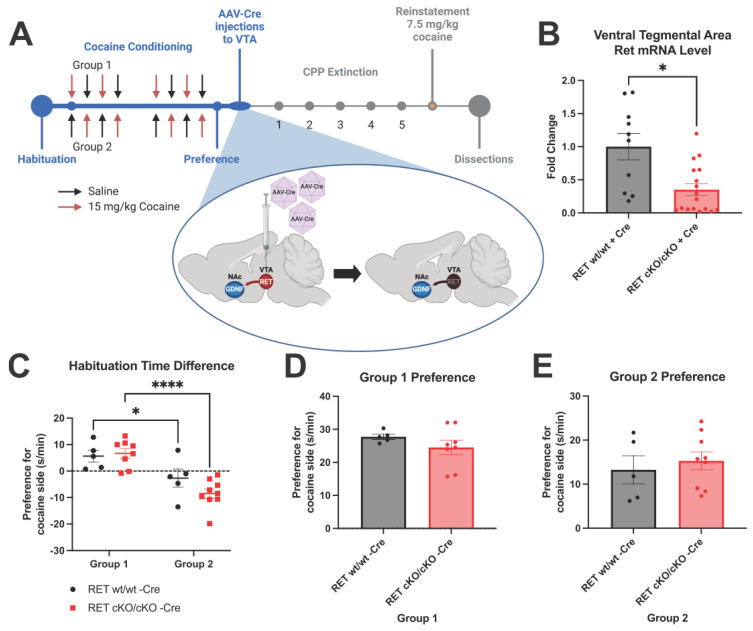
AAV-Cre-mediated reduction of VTA RET expression following cocaine-induced CPP. (**A**) Experimental scheme. Darkened RET in VTA after AAV-Cre injection represents reduction of endogenous RET expression. RET cKO mice were then evaluated 3, 5, 7, 9, and 11 days after AAV injections, followed by CPP reinstatement. Image created using biorender.com. (**B**) Ret mRNA levels measured after experiment show on average about 60 to 70% reduction in VTA (Welch’s unpaired *t*-test *p* < 0.05, animals per group = 10–17). (**C**) Place preference during habituation demonstrates a significant difference between group 1 and group 2 preferences for room to be paired with cocaine (2-way ANOVA significant difference between groups *p* < 0.0001, with Holm–Šídák post hoc test significant individual *p*-values depicted * = *p* < 0.05, **** = *p* < 0.0001, animals per group = 5–9). (**D**,**E**) No difference observed between genotypes in preference for cocaine side after cocaine-induced CPP prior to AAV-Cre-mediated reduction in mutants in either group 1 (**D**) or group 2 (**E**) (Welch’s unpaired *t*-test *p* > 0.05, animals per group = 5–9).

### 3.2. Animals with about 60–70% Reduction of VTA RET Have Hastened Extinction and Decreased Preference upon Cocaine Reinstatement

No significant differences in overall locomotor activity were observed between the genotypes during CPP testing, with differences only occurring as an effect of saline or cocaine treatment (Figure 2A). Notably, CPP extinction after the 60–70% reduction of VTA Ret was hastened in the mutant animals, with a significant 2-way repeated measures ANOVA interaction term (ANOVA interaction *p* < 0.05; Figure 2B). Further, preference for the cocaine side was significantly reduced in the mutants following CPP reinstatement with half of the original cocaine dose injected i.p. (unpaired *t*-test *p* < 0.05; Figure 2C). After the experiment, qPCR analysis of key plasticity and dopamine system-related genes in the VTA revealed no significant differences between wild type and mutants (Figure 2D), although there was a potential nonsignificant trend towards increased levels of Gdnf mRNA (*p* = 0.16).

### 3.3. GDNF cKO Mice upon AAV-Cre Delivery Have about 70–80% Reduction of GDNF from the NAc

The effects of GDNF on dopamine neurons are mediated by RET [34,35]. However, the GDNF-driven changes in dopamine metabolism occur mainly in the striatum where GDNF is expressed [38,40]. Thus, next we studied the effect of GDNF reduction in the NAc after cocaine CPP had been established. Here we injected AAV-Cre, this time into the NAc of GDNF cKO animals [49], to specifically target accumbal GDNF (Figure 3A). As shown on Figure 3B, Gdnf mRNA expression, which is known to correspond to GDNF protein level in mouse brain and other tissues [38,39,49,52] was successfully reduced by about 70–80% in the NAc via AAV-Cre injections. We next followed extinction over the next month before administering half the original dose of cocaine to reinstate preference (Figure 3A). Then we further confirmed our findings by again observing extinction after reinstatement before reinstating for a second time (Figure 3A). While habituation was unchanged between the groups (Figure 3C), group 1 displayed a significant preference difference *a priori* between the genotypes before AAV-Cre injections were performed (Figure 3D). Because this *a priori* preference difference could bias results and make interpretation of any effects due to reduction of GDNF difficult, group 1 was excluded from further analysis of CPP. Group 2 did not have significant preference differences observed *a priori* between the genotypes (Figure 3E), and thus served as experimental group for the behavioral analysis of CPP.

### 3.4. No Differences in Locomotor Activity Were Observed between the Mutants and Controls

Prior to AAV-Cre injections, animals of both groups displayed decreased locomotor activity after saline injections and increased locomotor activity due to cocaine, and, as expected, no differences between genotypes were observed (Figure 4A). During CPP extinction and reinstatement after AAV-Cre injections, no differences in locomotor activity between the genotypes were observed (Figure 4A). Half of the original cocaine dose used for the first reinstatement did also significantly increase locomotor activity compared to habituation, although no differences were observed between genotypes (Figure 4A).

### 3.5. Animals with about 70% Reduction in NAc GDNF Have Prolonged Extinction and Increased Preference upon Cocaine Reinstatement

After AAV-Cre injection, the mutant animals of group 2 showed markedly prolonged CPP extinction, with preference failing to reach control levels, even after 1 month (Figure 4B). Reinstatement of CPP with half the original dose of cocaine produced a significantly increased preference in the mutants (Figure 4C). Thus, GDNF reduction in NAc acted in a manner opposite to RET reduction in the VTA. To gain further data on this effect, we then measured extinction after cocaine reinstatement (Figure 3A). CPP after reinstatement remained increased in the mutants (Figure 4D), although no significant difference in preference was observed after the second reinstatement (Figure 4E). Together, these data indicate that a 70–80% reduction of GDNF in the NAc after the onset of cocaine addiction increases the rewarding effects of cocaine.

### 3.6. Analysis of mRNA Levels of Key Dopamine-Related and Plasticity Genes in the NAc and VTA after 70–80% GDNF Reduction in the NAc

Next, we addressed how NAc GDNF affects accumbal dopamine function by analyzing the expression level of dopamine-related and plasticity genes. We found that within the NAc, mRNA expression of proteins associated with dopamine biosynthesis and function, DAT, TH, DRD1, DRD2, and VMAT2, were largely unchanged, as were Ret and Gfra1 mRNA levels (Figure 5A). However, we found that mRNA levels encoding for BDNF, a known marker of neuroplasticity following cocaine exposure [53], was significantly upregulated in the NAc (Figure 5A). In line with the known RET and TH expression inducing effects of increased GDNF [38,39,54,55], we found reductions of RET, DAT, and TH encoding mRNA levels in the VTA after 70–80% GDNF reduction in the NAc (Figure 5B). To assess dopamine receptor expression in VTA neurons, we analyzed both Drd1 and Drd2 mRNA expression; however, no significant differences were observed (Figure 5B). Levels of Gfra1, Vmat2, and Bdnf mRNA in the VTA were also not significantly different between genotypes (Figure 5B). Taken together, our findings indicate that appropriate GDNF activity in NAc and dampened RET activity in the VTA facilitates recovery from cocaine addiction.

## 4. Discussion

Constitutively active RET has been shown to strongly prolong CPP extinction to amphetamine [44], suggesting that RET inhibition might be a plausible mechanism to facilitate the extinction of psychostimulant addiction. However, these MEN2B mice have constitutively upregulated RET throughout development which results in an almost two-fold increase in tissue dopamine and dopamine fiber density in the striatum and NAc as well as an increase in the number of dopamine neurons, at least in the substantia nigra [42]. Due to massive developmental alterations, it is impossible to predict the effect of adult-onset RET reduction on addiction. To conclude about potential therapeutic approaches involving this system, experiments addressing adult-onset reduction at the therapeutically relevant stage—after the onset of addiction—is instrumental. Here, through independent accumbal GDNF or VTA RET reduction after establishment of cocaine-induced conditioned place preference, we investigate this pathway’s potential to treat cocaine addiction.

We report that a 60–70% reduction of RET after the onset of cocaine addiction in the VTA hastens CPP extinction and reduces preference upon reinstatement compared to controls. Thus, partial inhibition of RET in the VTA may be beneficial in treating cocaine addiction. However, here we have targeted only the VTA for RET reduction. It is interesting to note that the difference between habituation and cocaine locomotor activity prior to AAV injection was not significant in the RET wt/wt animals, whereas it was significantly different in the RET cKO/cKO animals. This could indicate an initial difference in response to cocaine already inherent in the animals prior to RET reduction. However, it is most likely that this lack of significant effect in the wild type animals is due to the higher variation in the wt group. Importantly, there was no significant difference in locomotor activity observed between the RET wt/wt animals and the RET cKO/cKO animals when analyzed by *t*-test, indicating these groups are not inherently different. Future studies are required to determine whether constitutive adult-onset antagonism of RET would also be beneficial, or if it is specifically the reduction of RET levels in the dopamine neuron cell bodies in the VTA which has potential clinical benefit.

Importantly, we found that endogenous RET reduction in the VTA and endogenous GDNF reduction in the NAc have opposite effects on CPP extinction and reinstatement. While it is unlikely due to the low expression of neurturin or GFRa2 in the NAc [24,56], neurturin may potentially activate RET independent of GDNF-GFRa1 signaling and have effects on dopamine neurons [57], perhaps partially explaining this difference. Still, such a bidirectional or spatially segregated effect of VTA RET and NAc GDNF on cocaine-induced CPP is, in fact, in line with previous data which utilized ectopic overexpression. Namely, previous studies report that a striatal chronic ectopic GDNF increase reduces reward responses to cocaine [45] and to alcohol [46]. Additionally, intrastriatal and intra-accumbal transplantation of GDNF-producing astrocytes or nanoparticles reduces cocaine self-administration [58,59]. On the other hand, RET constitutive activation increases reward responses to psychostimulants [42,44]. Further evidence for spatially segregated functions of GDNF-RET signaling in cocaine reward comes from a study which demonstrated that an acute, high dose of GDNF injected into the VTA enhances cocaine-seeking behavior [60]. This suggests that increased GDNF-RET signaling in the VTA enhances and the reduction in VTA RET levels reduces cocaine addiction. Notably, this effect of excess GDNF delivery into the VTA was blocked by an ERK-inhibitor, downstream of RET signaling [60].

It is interesting that differences in behavioral response to cocaine administration were not observed between genotypes, as locomotor activity was not significantly different between genotypes. This suggests a specific role of NAc GDNF and VTA RET signaling in cocaine addiction, but not cocaine-induced motor activity. Indeed, there is a nonsignificant trend towards an increase in locomotor activity in the RET cKO animals with about 70% reduction of RET levels during reinstatement despite having reduced cocaine preference. Together, this could indicate the possibility of a specific role of GDNF-RET signaling in the cocaine reward.

We also observed an upregulation of BDNF in the NAc of GDNF cKO animals, which is potentially of particular relevance to the observed prolonged CPP extinction. Namely, BDNF and TrkB knockdown in mouse NAc has been shown to reduce CPP [61,62,63,64] and, conversely, BDNF overexpression in NAc has been shown to increase CPP and delayed extinction suggesting that increased BDNF at least in part mediates the effect of GDNF reduction [65,66]. Further supporting this idea, the minipump infusion of BDNF into the NAc has been shown to enhance responding for conditioned reinforced stimuli, much like CPP [67]. This mechanism, at least in part, could account for the prolonged CPP extinction observed in our animals with reduced NAc GDNF.

We also found that after cocaine administration, a 70–80% reduction of GDNF in the NAc moderately reduces DAT-, RET-, and TH-encoding mRNA expression in the VTA. The reduction in dopamine-related gene expression could also be due to histological alterations in midbrain dopamine neurons such as fiber or cell degeneration. However, degeneration is unlikely as GDNF has been demonstrated to not be required for the survival of midbrain dopaminergic neurons in vivo in GDNF knockout animals [49]. Still, future studies should indicate whether there are histological alterations of the NAc-projecting dopamine neurons after cocaine administration. Nevertheless, the data do suggest an overall reduction in endogenous dopamine in response to cocaine upon GDNF reduction, which at first sight may seem surprising given dopamine levels have been shown to positively correlate with CPP [42,68]. However, a recent study showed, in contrast to dopamine levels in the dorsomedial striatum, a negative correlation between NAc dopamine levels and active lever responding during the first day of conditioned reward extinction [16]. Together with our results, this may indicate that a modest reduction in DA predisposes one to a slower cocaine-induced CPP extinction.

GDNF itself has for decades been known to have potent effects on dopamine system function and survival in models of Parkinson’s disease [20,21]. Notably, both developmental GDNF conditional knockout [40] and developmental upregulation of endogenous GDNF expression increase DAT activity in the striatum [38,39]. In addition, the upregulation of endogenous GDNF expression increases local striatal dopamine levels, release, and reuptake, while simultaneously dampening DA and DAT levels in the prefrontal cortex where GDNF is not expressed at high levels [24,38], indicating that GDNF exerts its effects on the dopamine system locally. Indeed, upon GDNF binding, RET can activate MAP kinases locally at the nerve fibers or can be internalized into clathrin-coated vesicles and retrogradely transported long distances into alternative cellular compartments in the dopamine cell bodies in the VTA [28,69,70,71,72].

It is intriguing that a reduction of GDNF levels in the NAc of GDNF cKO animals also resulted in a modest reduction in RET mRNA levels in the VTA. Given that the reduction in RET mRNA levels in the GDNF cKO is associated with prolonged CPP, whereas the reduction of RET alone in the VTA hastened CPP extinction, this could indicate that a reduction of RET expression may not have a significant or specific role in decreasing cocaine-induced CPP, or, alternatively, that maintaining or amplifying NAc GDNF is necessary in addition to reducing VTA RET expression to hasten cocaine-induced CPP extinction. It is interesting to note the potential nonsignificant trend towards increased GDNF in the VTA of the RET cKO animals. This indicates support for the latter hypothesis. However, future studies are required to adequately resolve this interesting question.

## 5. Conclusions

Clinically, pharmacotherapeutic management of cocaine addiction would require treatment to be applied after cocaine use disorder has already been established to attain an initial period of abstinence and assist in preventing relapse [73]. Here, we demonstrate that conditional reduction of RET from the VTA results in faster CPP extinction and reduced reinstatement after subsequent application of cocaine (Figure 6, left). The opposite effect was observed when GDNF was reduced in the NAc (Figure 6, right). Our results imply that, in theory, the application of RET blockers which preferentially block internalized RET signaling in the VTA cell bodies could serve as future treatment of cocaine addiction. Potentially this could be combined with small molecule GDNF mimetics which mainly enhance NAc GDNF function. Future research should validate the validity of targeting GDNF-RET signaling in other addiction models and reveal if such pharmacological splitting of GDNF-RET signaling in the NAc and VTA is possible.

## Figures and Tables

**Figure 2 biomolecules-13-00761-f002:**
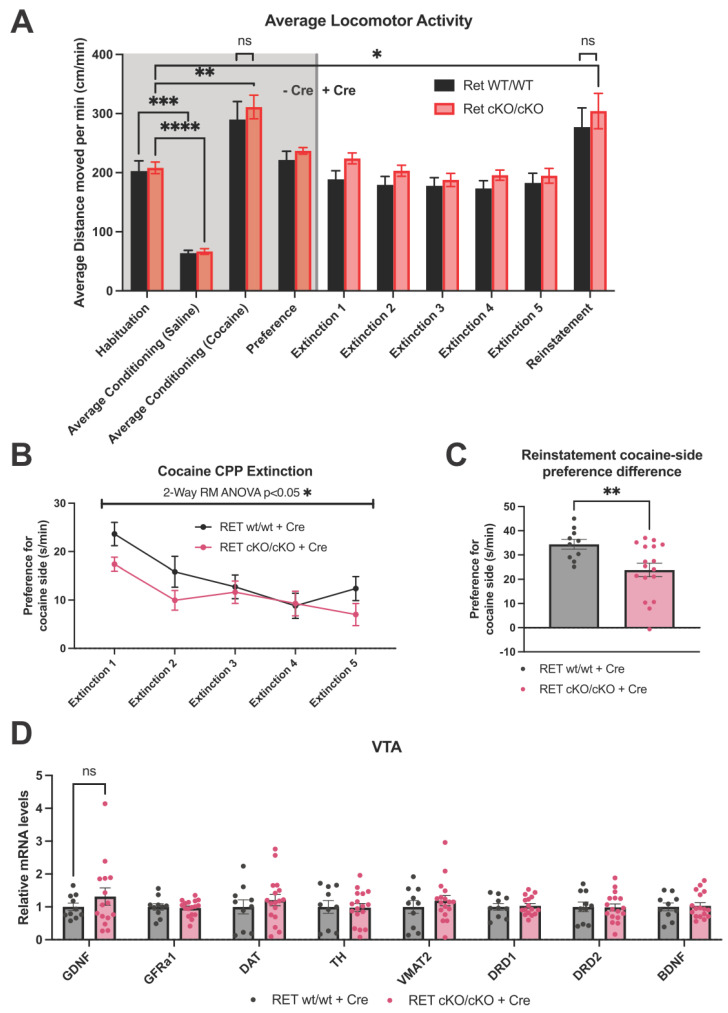
Cocaine reward in RET cKO animals after reduction of VTA RET expression. (**A**) Locomotor activity during CPP experiments was not different between the genotypes but was affected by cocaine administration (repeated measures mixed-effects analysis significant time effect *p* < 0.0001, with Holm–Šídák post hoc test significant comparisons depicted on graph * = *p* < 0.05; ** = *p* < 0.01; *** = *p* < 0.001; **** = *p* < 0.0001, animals per group = 10–17). (**B**) Preference for cocaine side extinction after cocaine-induced CPP (measured as time spent on cocaine side-group average time spent on saline side) is hastened in RET cKO animals with approx. 60 to 70% reduced VTA Ret mRNA (repeated measures 2-way ANOVA significant interaction term *p* < 0.05, animals per group = 10–17). (**C**) Cocaine-induced CPP reinstatement using half the original dose of cocaine is significantly reduced in mutant animals (Welch’s unpaired *t*-test *p* < 0.05, animals per group = 10–17). (**D**) No significant alterations in the mRNA levels of key genes in the VTA were observed following cocaine exposure (multiple *t*-tests followed by Benjamini, Krieger, and Yekutieli two-stage set-up method for controlling the false discovery rate; Q = 5%, animals per group = 10–17).

**Figure 3 biomolecules-13-00761-f003:**
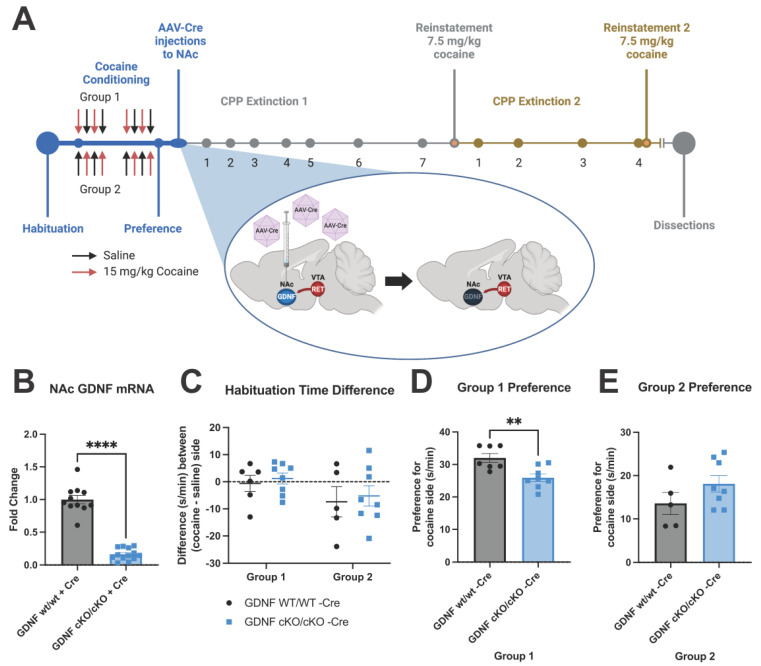
AAV-Cre-mediated reduction of NAc GDNF expression following cocaine-induced CPP. (**A**) Experimental scheme. Darkened GDNF in NAc after AAV-Cre injection represents reduction of endogenous GDNF expression. GDNF cKO mice were then evaluated for CPP extinction 4, 7, 10, 14, 17, 23, and 30 days after AAV injections. CPP reinstatement was evaluated 34 days after AAV injections. After reinstatement, CPP extinction was again evaluated 36, 41, 49, and 56 days after the original AAV injections, with a second reinstatement evaluated on day 57. Image created using biorender.com. (**B**) Significant approx. 70 to 80% reduction of Gdnf mRNA was observed in the NAc in mutant animals following experiments (Welch’s unpaired *t*-test **** = *p* < 0.0001, animals per group = 11–13). (**C**) No significant difference between groups was observed in preference for either side during habituation (2-way ANOVA *p* > 0.05, animals per group = 5–8). (**D**) During cocaine-induced CPP, prior to AAV-Cre injections, Group 1 showed a random, *a priori* significant difference in preference for cocaine side between the genotypes (Welch’s unpaired *t*-test ** = *p* < 0.01, animals per group = 7–8). (**E**) No difference between genotypes was observed in cocaine-induced CPP in group 2 (Welch’s unpaired *t*-test *p* > 0.05, animals per group = 5–8).

**Figure 4 biomolecules-13-00761-f004:**
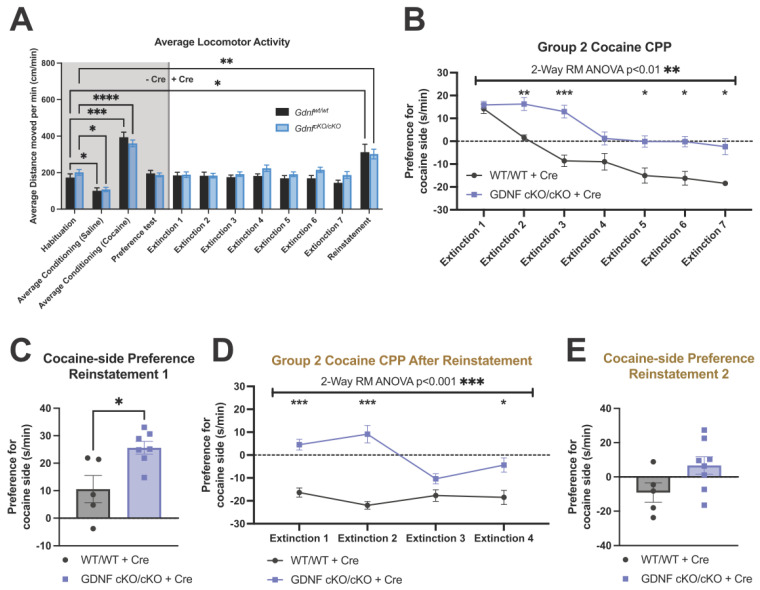
Cocaine reward in GDNF cKO animals after reduction of NAc GDNF expression. (**A**) Locomotor activity during first half of CPP experiments was not different between the genotypes but was affected by cocaine administration (repeated measures mixed-effects analysis significant time effect *p* < 0.0001, with Holm–Šídák post hoc test significant comparisons depicted on graph * = *p* < 0.05; ** = *p* < 0.01; *** = *p* < 0.001; **** = *p* < 0.0001, animals per group = 12–16). (**B**) Preference for cocaine side after cocaine-induced CPP (measured as time spent on cocaine side-group average time spent on saline side) is prolonged in GDNF cKO animals with ~70 to 80% reduced NAc Gdnf mRNA (repeated measures 2-way ANOVA significant interaction term *p* < 0.01, with Holm–Šídák post hoc test significant comparisons at individual time points depicted on graph * = *p* < 0.05; ** = *p* < 0.01; *** = *p* < 0.001; **** = *p* < 0.0001, animals per group = 5–8). (**C**) Cocaine-induced CPP reinstatement using half the original dose of cocaine is significantly increased in mutant animals (Welch’s unpaired *t*-test *p* < 0.05, animals per group = 5–7, one outlier was removed in mutants justified by Grubbs’ test). (**D**) After the first reinstatement, cocaine-induced CPP is again increased in the mutant animals during extinction (repeated measures 2-way ANOVA significant interaction *p* < 0.001, with Holm–Šídák post hoc test significant comparisons at individual time points depicted on graph * = *p* < 0.05; ** = *p* < 0.01; *** = *p* < 0.001; **** = *p* < 0.0001, animals per group = 5–8). (**E**) The second cocaine-induced CPP reinstatement again with 7.5 mg/kg cocaine was not any more significant between genotypes (Welch’s unpaired *t*-test *p* < 0.05, animals per group = 5–8).

**Figure 5 biomolecules-13-00761-f005:**
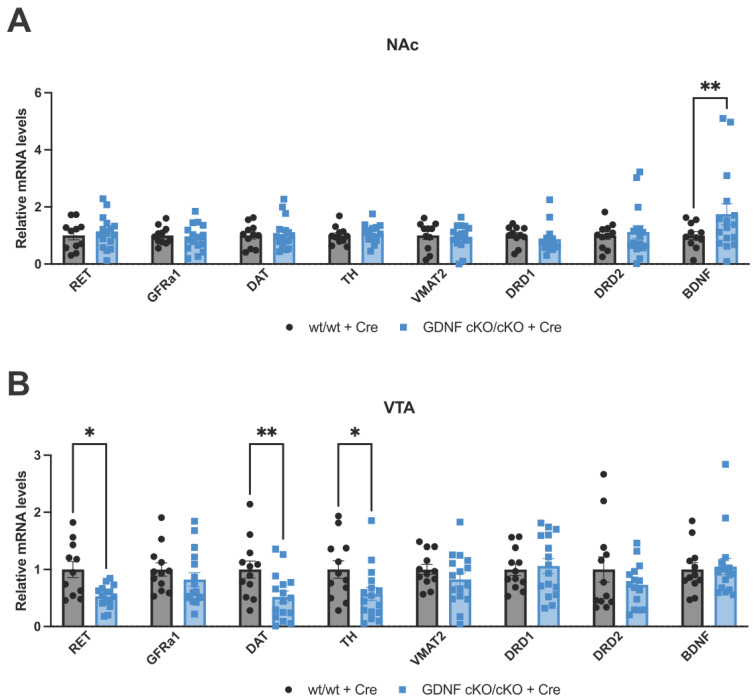
Dopamine-related and plasticity gene expression in the NAc and VTA following cocaine exposure in animals with reduced GDNF. (**A**) Key gene mRNA levels in the NAc reveal a significant increase in Bdnf mRNA after cocaine exposure in GDNF cKO animals with ~70% reduction of Gdnf mRNA (multiple *t*-tests followed by the Benjamini, Krieger, and Yekutieli two-stage set-up method for controlling the false discovery rate; *t*-test *p*-values of significantly different “discoveries” are depicted on graph * = *p* < 0.05; ** = *p* < 0.01, animals per group = 11–16). (**B**) Dat, Th, and Ret mRNA levels are significantly reduced in the VTA following cocaine exposure in GDNF cKO animals with ~70% reduced Gdnf mRNA (multiple *t*-tests followed by the Benjamini, Krieger, and Yekutieli two-stage set-up method for controlling the false discovery rate; *t*-test *p*-values of significantly different “discoveries” are depicted on graph * = *p* < 0.05; ** = *p* < 0.01, animals per group = 12–16).

**Figure 6 biomolecules-13-00761-f006:**
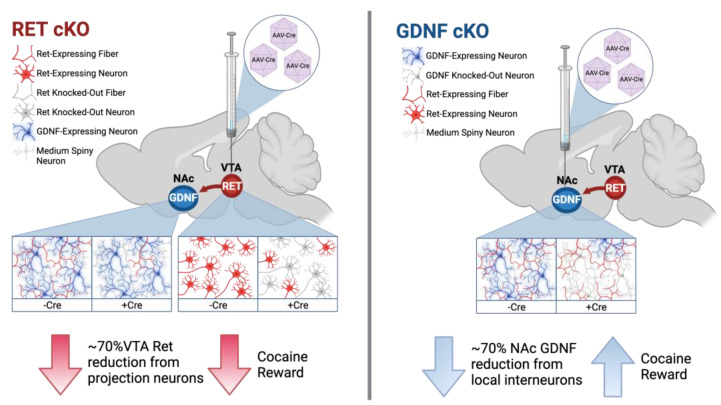
Bidirectional, spatially segregated modulation of cocaine reward via GDNF and RET. Injection of AAV-Cre into the VTA to reduce RET expression in cKO animals (**left**) reduces cocaine reward, while injection of AAV-Cre into the NAc to reduce GDNF in cKO animals (**right**) increases reward. Created with biorender.com.

**Table 1 biomolecules-13-00761-t001:** Primers used for genotyping.

Primer	Forward Sequence	Reverse Sequence
Ret cKO	ACT CCT TGG GCC TGC TGA G	GAG GCA GGA AGG CCT GTG
Gdnf cKO	5′: CTC ATT TCC CAC AGG GAA CTG 3′: GAA ACC AAG GAG GAA CTG ATC	3′: TCT TCT GCC TCT GCC TCC G

**Table 2 biomolecules-13-00761-t002:** Primers used for RT-qPCR analysis.

Primer	Forward Sequence	Reverse Sequence
mRet	TCC CTT CCA CAT GGA TTG A	ATC GGC TCT CGT GAG TGG TA
mGDNF	CGC TGA CCA GTG ACT CCA ATA TGC	TGC CGC TTG TTT ATC TGG TGA CC
mGfra1	TTC CCA CAC ACG TTT TAC CA	GCC CGA TAC ATT GGA TTT CA
mTh	CCC AAG GGC TTC AGA AGA G	GGG CAT CCT CGA TGA GAC T
mVmat2	ATG CTG CTC ACC GTC GTA GT	TTT TTC TCG TGC TTA ATG CTG T
mDat	AAC CTG TAC TGG CGG CTA TG	GCT GAC CAC GAC CAC TAC A
mDrd2	ACA CAC CGT ACA GCT CCA AG	GGA GTA GAC GAC CAC GAA GGC AG
mDrd1	GCG TGG TCT CCC AGA TCG	GCA TTT CTC CTT CAA GCC CCT
mBdnf	GGC CCA ACG AAG AAA ACC AT	AGC ATC ACC CGG GAA GTG T
mbActin	CTA AGG CCA ACC CTG AAA AG	ACC AGA GGC ATA CAG GGA CA

## Data Availability

All data related to this manuscript are available upon request.

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
