# Peer review of "Opposing Spatially Segregated Function of Endogenous GDNF-RET Signaling in Cocaine Addiction"

_biomolecules, 2023, doi:10.3390/biom13050761_

Round 1

Reviewer 1 Report

The work of Garton et al. was able to differentiate the effect of the conditional knockout (cKO) of two components of a signaling pathway, namely GDNF-RET signaling, in different areas of the dopaminergic pathway on cocaine addiction through CPP. cKO of RET improved cocaine CPP extinction and lowered CPP relapse, while GDNF cKO resulted in the opposite. With both being expressed in brain regions with dopaminergic influence, it is intriguing how the cKO of these targets lead to divergent consequences. They were also able to determine mRNA alterations in the NAC and VTA following cocaine CPP in GDNF cKO mice, which may support addiction-related dopaminergic alterations. While the objective of their study is clearly stated, and the concept itself is scientifically significant in recent times, there are several areas where they may improve their work, along with some queries that need to be clarified. Overall, their manuscript gave the impression that RET might be a potential target for reducing established cocaine addiction, however their discussion focused more on GDNF.

Major comments:

·         In lines 49 to 60, the authors mentioned the expression location of GDNF and RET, which are found on dopaminergic neurons, thus potentially regulating reward and addiction. The authors may expound more on this mechanism, for example, how increasing GDNF or RET expression actually modulates dopaminergic activity (the downstream cascade involved), aside from stating the effects of their expression level on addiction-related behavior.

·         Was it indicated in Figure 2A if conditioned RET wt/wt mice have significantly higher distance moved than habituation, similar to the comparison done in Figure 4A? If not, this may indicate an inherent difference regarding cocaine sensitivity among mice, even prior to AAV-Cre injections. Comparison of the locomotor activity between wt and cKO mice may also support significant differences found during extinction and reinstatement, and thus should also be indicated.

·         It is intriguing how differences in CPP extinction and reinstatement were found between wt and cKO mice, but not in their locomotor activity, despite cocaine being a known psychostimulant. Moreover, the locomotor activity of RET cKO mice is seemingly higher than that of RET wt during reinstatement, despite having lower cocaine preference. This may indicate a specificity of this pathway (GDNF-RET signaling) to may only govern cocaine reward, but not cocaine-induced stimulation. The authors may supplement more to this in the Discussion section.

·         What was the reason for only determining dopamine-related genes in GDNF cKO mice? Such targets should also be checked in RET mutant mice. It might be presumed that their disparate behavior may be reflected by divergent alterations in the same set of genes, thus supporting the role of this pathway in cocaine addiction. Also, why was Drd1 mRNA level not included?

·         RT-PCR results revealed that GDNF cKO resulted also in a significant reduction of RET mRNA levels in the VTA. However, given the results showing that RET cKO reduced cocaine CPP reinstatement, RET reduction might not have a significant or specific role on cocaine addiction. The authors should discuss this potentially contradicting finding, and thus GDNF mRNA levels in RET cKO mice should also be determined.

Minor comments:

·         In the abstract, the flow of experiments implies that GDNF cKO experiments were performed before RET cKO experiments. However, the logic of the data arrangement implies the contrary. Rephrasing the abstract might be more appropriate for easier and accurate understanding.

·         The mRNA changes found related to the dopaminergic system should also be mentioned in the abstract.

·         The number of samples in each group should also be indicated in the figure legends.

·         Line 81: future development “of” GDNF-RET…

·         The authors may also discuss a probable cause for the habituation time difference between group 1 and 2 of RET cKO/cKO-Cre mice. Were experimental conditions different?

·         What was the particular reason for using ActinB with GDNF cKO and the geometric mean of the others with RET cKO as reference genes?

·         The order of the presentation of the methodology should follow the logical flow of the experiment. Experiments requiring brain tissue samples must be stated after brain dissection/extraction.

·         Line 177: “post-hoc”

·         Statement of statistical significance should be consistent, whether P or Q values are indicated, although P values are more common. Line 207 even indicated “p < 0.5”.

·         Please have consistency with the titles of subheadings, for example, 3.1 and 3.2.

·         Figure 3C: “GDNF” WT/WT-Cre

Author Response

Please see the attachment, our responses are in blue.

Reviewer 2 Report

The authors explored the neurotrophic role of the GDNF/RET system in cocaine addiction using the tissue-specific KO mice. The authors mainly evaluated their effects on scores of conditioned place preference tests. The authors found that GDNF reduction prolongs the place preference whereas RET decline promotes its extinction. Thus, RET would be a potential drug target for its medication.

The experiments are well designed and nicely controlled. The conclusion is unique and noteworthy. However, several issues need to be improved before publication.

One of the major problems of this MS is the quality of English. Although the reviewer is not a native English speaker, there are many awkward  expressions even in ABSTRACT; crucial to the establishment, NAc projecting dopamine neurons in VTA.

1. Spell out GDNF in the abstract.

2. Provide more detailed explanations of Group1 and Group 2.

3. The use of FDR statistics is rather confusing with the given low number of comparisons. Simple T-test is fine with the proper explanation.  

4. Please use Welch T-test with the given group difference of SD.

5. GDNF is not a sole ligand for RET. Please discuss the contribution of other RET ligands for this finding.

6. There is no histological examination in this study. The mRNA changes in Fig 5 might stem from DA fiber or cell degeneration but not from phenotypic mRNA changes. 

7. In FIg5, what is the number 0.024080 and why it is same in all??? 

Author Response

(The authors gave the same response as above.)

Reviewer 3 Report

The very well conceived and structured research is complex, but rigorously consequential. The figures are complicated but self explaining. The references are adequate.

lines 26-28 and 398-400. This statement is an interesting hypothesis of the authors, but obviously it has not yet been verified in other models and therefore it cannot be transferred to clinical practice, a step that will still require a lot of work. This concept, which is present in the following sentence, should be underlined and expanded.

Author Response

(The authors gave the same response as above.)

Round 2

Reviewer 1 Report

I commend the authors for their hard work, as they have considered the recommendations and answered most of the queries. Indeed, the quality of their work has improved. However, before being acceptable for publication, there are a few minor points that need to be clarified.

1. Line 409: This statement must not conclude that GDNF-RET is limited to cocaine reward. This finding should only convey a possibility, as it may also influence other cocaine effects (reinforcing/withdrawal). 

2. I commend the authors for updating their qPCR data and including dopamine-related targets also in RET cKO mice VTA, along with Drd1. However, it is unclear why they checked Drd1 in the VTA and not the NAC. In terms of addiction, manifestation of reward/addiction usually implicate DRD1 that are expressed in dopaminergic projections, such as the frontal cortex and NAC. Please provide a reason why Drd1 was not investigated in the NAC.

3. Line 212: "p < 0.5" indicates no significant differences. Please rectify.
